# The Evolution, Expression Patterns, and Domestication Selection Analysis of the Annexin Gene Family in the Barley Pan-Genome

**DOI:** 10.3390/ijms25073883

**Published:** 2024-03-30

**Authors:** Liqin Chen, Kunxiang Chen, Xi Xi, Xianghong Du, Xinyi Zou, Yujia Ma, Yingying Song, Changquan Luo, Song Weining

**Affiliations:** 1State Key Laboratory of Crop Stress Biology in Arid Areas, College of Agronomy, Northwest A&F Univesity, Xianyang 712100, China; liqinchen@nwafu.edu.cn (L.C.); chenkunxiang2022@163.com (K.C.); xixi_@nwafu.edu.cn (X.X.); 2College of Agronomy, Northwest A&F University, Xianyang 712100, China; xianghongdu@nwsuaf.edu.cn (X.D.); celine_rain@163.com (X.Z.); 3College of Landscape Architecture and Art, Northwest A&F University, Xianyang 712100, China; 2022011423@nwsuaf.edu.cn; 4College of Plant Protection, Northwest A&F University, Xianyang 712100, China; syy13373770191@163.com; 5College of Life Sciences, Northwest A&F University, Xianyang 712100, China; 2023014268@nwsuaf.edu.cn

**Keywords:** barley pan-genome, annexin, expression profiles, domestication selection

## Abstract

Plant annexins constitute a conserved protein family that plays crucial roles in regulating plant growth and development, as well as in responses to both biotic and abiotic stresses. In this study, a total of 144 annexin genes were identified in the barley pan-genome, comprising 12 reference genomes, including cultivated barley, landraces, and wild barley. Their chromosomal locations, physical–chemical characteristics, gene structures, conserved domains, and subcellular localizations were systematically analyzed to reveal the certain differences between wild and cultivated populations. Through a cis-acting element analysis, co-expression network, and large-scale transcriptome analysis, their involvement in growth, development, and responses to various stressors was highlighted. It is worth noting that HvMOREXann5 is only expressed in pistils and anthers, indicating its crucial role in reproductive development. Based on the resequencing data from 282 barley accessions worldwide, genetic variations in thefamily were investigated, and the results showed that 5 out of the 12 identified HvMOREXanns were affected by selection pressure. Genetic diversity and haplotype frequency showed notable reductions between wild and domesticated barley, suggesting that a genetic bottleneck occurred on the annexin family during the barley domestication process. Finally, qRT-PCR analysis confirmed the up-regulation of *HvMOREXann7* under drought stress, along with significant differences between wild accessions and varieties. This study provides some insights into the genome organization and genetic characteristics of the annexin gene family in barley at the pan-genome level, which will contribute to better understanding its evolution and function in barley and other crops.

## 1. Introduction

Barley (*Hordeum vulgare* L.) is an ancient domesticated annual plant, initially identified at a Natufian burial site in Raqefet Cave, Israel (13,700–11,700 cal. BP) [1]. Globally, it holds a significant position as one of the major cereal crops, with yields surpassed only by wheat, corn, and rice [2]. Originally cultivated for human consumption, barley has evolved into a primary grain for feed, malting, and brewing [3]. Among cereals, barley stands out as the most adaptable, showcasing a remarkable adaptability, with cultivation extending from the arctic to subtropical zones. Hordeum species are widespread in areas with a Mediterranean climate, and this genus is also found in regions with both oceanic and continental climates [4]. Furthermore, barley’s rich genetic background enhances its significance. The relatively straightforward diploid genetics of barley and its close relationship to other crops in the Triticeae family within the grass family Poaceae facilitate the transfer of research knowledge from barley to other cereal crops, such as wheat and rye [5].

Extreme weather events, rising temperatures, and insufficient precipitation, among other climate changes, impact agricultural production. This may lead to droughts, floods, and other natural disasters, damaging crop yields and posing a threat to food security [6]. Researching the genetic mechanisms of crops to resist drought and tolerate salt, cold, heat, and other adverse conditions is one of the most cost-effective and sustainable solutions to enhancing crop productivity and yield stability, contributing significantly to the global food security agenda [7,8]. In response to the global emergence of extreme climates leading to natural disasters, substantial progress has been made over the past two decades in the study of plant annexins, concerning their roles in growth, development, and responses to environmental stresses [9,10]. In the evolutionary process, plant annexins show high conservation, and the ancestral sequences of angiosperms can be traced back to unicellular green algae and a non-vascular land plant species [11]. Plant annexins are expressed in a variety of plant tissues, playing a significant role in regulating diverse biochemical and morphological processes, as well as influencing plant growth, development, and responses to both biotic and abiotic environmental stimuli [10,12]. During the elongation stages of cotton fiber differentiation, the highly expressed cotton annexin gene, GhAnn2, plays a pivotal regulatory role by modulating Ca^2+^ signaling flux in fiber development [13]. ANN1 and ANN2 play crucial roles in facilitating post-phloem sugar transport to the root tip, indirectly influencing photosynthetic rates in cotyledons [14]. In wheat and Arabidopsis, it has been demonstrated that annexin is expressed in the reproductive organs, influencing both pollen fertility and embryo development [15,16]. Annexin’s involvement in osmotic stress and ABA signaling, as well as its demonstrated role in drought stress and salt stress, has been confirmed in multiple species. The overexpression of RsANN1a improved plants’ growth and heat tolerance [17]. A cotton phosphatase GhDsPTP3a and an annexin protein GhANN8b interact and reversely modulate Ca^2+^ and Na + fluxes in cotton salinity responses [18]. The Ca^2+^-permeable transporter ANNEXIN1 (AtANN1) mediates cold-triggered Ca^2+^ influx and freezing tolerance in Arabidopsis thaliana [19].

Pan-genomics is a comprehensive approach that captures the vast genetic diversity within a species or population. It has emerged as a robust tool for exploring genomic evolution, unraveling the origins and domestication processes of species and offering valuable insights for enhancing plant traits through genetic improvements [20]. The completion of the barley pan-genome contributes to tracing the evolutionary history and domestication processes of barley and its close relatives. It aids in breeding varieties better adapted to diverse environmental conditions, thereby promoting sustainable agriculture [21]. The objective of this study is to systematically identify the barley annexin gene family through genome-wide classification, and to recognize its gene structure, conserved domains, cis-regulatory elements, and phylogenetic classification. Additionally, the study systematically investigates its tissue expression patterns under normal conditions and gene expression under various abiotic stress conditions. Through a co-expression network analysis and GO enrichment analysis, the regulatory networks and biological processes in which it participates are studied. The domestication selection of annexin genes in barley and major haplotype differences are analyzed using large-scale population data. Finally, a quantitative real-time PCR (qRT-PCR) analysis is used to validate their transcriptional status under drought stress conditions, as well as differences in expression between wild and cultivated individuals.

## 2. Results

### 2.1. Identification of Annexin Genes in the Barley Pan-Genome

Both an HMMER search and the Blastp program were used for identification. A total of 144 putative annexin genes were identified in the 12 barley genomes (Appendix A). The number of each accession is shown in Table 1. The removal of annexins lacking intact conserved domains was conducted. As a result, there was no difference in the numbers of annexin genes among cultivars, landraces, and wild barley. They typically possess 12 annexin genes similar to those found in rye [22]. A chromosome distribution analysis revealed that the 12 Hvanns were not evenly distributed across all barley chromosomes (Appendix A). Specifically, chr2H, chr4H, and chr5H each contained two Hvanns members, chr3H and chr6H harbored two Hvanns members, and chr1H had the highest number of Hvann members at three. Additionally, it is worth noting that two wild types, EC-N1 and EC-S1, exhibited distinct variations. Specifically, chr3H in EC-N1 did not possess any Hvanns, while EC-S1 had two Hvanns.

According to this analysis, the physicochemical properties of the annexin genes are shown in Appendix A. The annexin genes encoded proteins ranging from 68 (HvS1ann7) to 382 (HvS1ann9) amino acids in length, with isoelectric points (pIs) ranging from 5.51 (HvB1Kann5) to 11.45 (HvN1ann4) and molecular weights (MWs) varying from 7.63 (HvS1ann6) to 41.97 kDa (HvN1ann12). The GRAVY values ranged from −1.229 (HvN1ann4) to −0.115 (HvS1ann6), with an average of −0.39. All of the Hvanns proteins had a negative GRAVY score, indicating that the Hvanns proteins were hydrophobic. Furthermore, the results of subcellular localization prediction indicated that 51% of Hvanns were localized in the cytoplasm, while 25% were localized in the chloroplast. Additionally, some Hvanns were found to be distributed in the nucleus and mitochondria, with 7% of Hvanns showing a simultaneous presence in both the nucleus and cytoplasm (Table 1). These findings suggest that these proteins have diverse subcellular localizations, which are crucial for understanding their functions and interactions. It is evident that the physicochemical properties and subcellular localizations of annexin genes exhibit significant variations among wild genomes, while these differences are less pronounced among cultivars and landraces (Table 1).

### 2.2. Phylogenetic and Molecular Evolution Analysis of Barley Annexin Genes

To explore the phylogenetic relationships of barley annexin genes in barley genomes, a Maximum Likelihood (ML) model tree was constructed using proteins from 144 identified annexin genes, alongside 10 annexin genes from Oryza sativa and 35 annexin genes from Triticum aestivum (Figure 1). A gene structure analysis, conserved motif analysis, and phylogenetic tree analysis were performed to classify the annexin genes. Consistent with previous research on wheat and rye [16,22], the annexin genes identified in this study can be categorized into six distinct groups, each representing a specific protein family. The distribution of proteins among these subfamilies varies significantly, with Subfamilies I and III comprising 46–48 members, Subfamilies II and V having 31–32 members, and Subfamilies IV and VI containing 16 members each (Figure 1). Each group includes genes from the same chromosomes of barley, wheat, and rice, indicating a high conservation and consistency of Hvanns.

Regularities can be observed among the gene structures, conserved motifs, phylogenetic relationships, and subcellular localizations (Appendix A). Notably, a subgroup within group II of Hvanns is characterized by the presence of seven motifs and three annexin domains, while most members of group V possess four annexin domains and exhibit the longest intron structure (Appendix A). Hvann6 of group VI is exclusively found in the chloroplast (Appendix A). More than half of the genes in this family contain four annexin repeats, while a few, such as HvS1ann6 and HvN1ann4, have only one annexin repeat (Appendix A). Wild relatives are often located at the outermost branches of the tree, primarily due to their ancestral status of the cultivated type, showcasing higher levels of diversity compared to other genes within the same group. Conversely, there is relatively little variation between domesticated and cultivated varieties. In general, members clustered within the same clade based on their phylogenetic relationships exhibit analogous motif compositions, exon–intron structures, and cellular locations, implying potential similarities in their biological functionalities.

### 2.3. Cis-Element Analysis of Hvanns

In the investigation of gene regions and the 2 kbp upstream sequences of the genes, the online tool PlantCARE database was employed for prediction, revealing a total of 47 distinct cis-elements within the 2000 bp upstream region from the transcription start sites of the Hvanns. These cis-elements are extensively involved in stress responses, hormone responses, and metabolic regulation, as well as growth and development processes (Figure 2, Appendix A). Among them, the cis-acting regulatory elements G-box and ABRE, involved in light and abscisic acid responsiveness stress responses, were identified in almost all Hvanns. In particular, more than 11 copies of elements, such as Hv3365ann5, HvMOREXann5, and HvS1ann4, were found in these two elements. Additionally, this group of genes recognized over 11 copies of the MYB binding site (MBS), which is involved in drought inducibility. This result is consistent with our existing research, as the annexin gene has been found to be associated with drought tolerance in rice [23], durum wheat [24], Arabidopsis [25], and tomato [26].

In the realm of metabolic response processes, the O2-site, a cis-acting regulatory element intricately involved in the regulation of zein metabolism, was solely successfully identified; in wild, landrace, and variety barley, there were 10, 15, and 13 genes, respectively, containing this element, with domesticated barley having the highest number of genes containing this element (Appendix A). In 12 genomes, the AACA_motif element, which is involved in endosperm-specific negative expression, was found only in HvMOREXann6, Hv13821ann6, HvB1Kann6, and Hv3365ann6. The AACA_motif element has been shown to confer endosperm-specific expression of the rice storage protein glutelin gene GluA-3 [27]. This result suggests an association between annexin and seed storage. In all the cis-acting regulatory elements, the most common category was stress response, which suggests that it plays a crucial role in various stress responses. The presence of multiple copies of stress response elements in more than half of the Hvanns further supports its importance in different stress conditions. This information adds to our understanding of the regulatory mechanisms involved in the protein’s function during stress adaptation.

### 2.4. Expression Profiles of Hvanns in Tissues and under Stress Conditions Based on RNA-Seq Data

From a transcriptome perspective, the small number of Hvanns genes, yet their significantly varied transcriptional characteristics, implies a diversity of annexin functions. It is evident that HvMOREXann1 and HvMOREXann4 are predominantly expressed in the roots (Figure 3A), with HvMOREXann1 associated with metal ion AL stress (Figure 3B). This discovery is related to the involvement of Arabidopsis annexins in early seedling growth and development, particularly in the Golgi-mediated secretion process [25,28]. HvMOREXann7 and HvMOREXann6 exhibit distinct high expression levels in embryos isolated from 4-day-old germinating grains (Figure 3A). Furthermore, HvMOREXann7 is expressed in lodicules post-fertilization (Appendix A), suggesting its potential role in regulating normal embryo development, tissue formation, and influencing cell differentiation and organ formation in subsequent developmental processes. HvMOREXann10 is predominantly detected in the rachis, stem, and anthers (Figure 3A and Appendix A). Nevertheless, there is limited research on the role of annexins in the cell wall and secondary cell wall. Notably, among the 16 representative tissues in PRJEB14349, HvMOREXann5 is not included. This gene is solely expressed in pistils and anthers (Figure 3A and Appendix A), akin to annexin 5 in Arabidopsis, which plays a key role in reproductive development and is vital for pollen and embryo formation [15].

In addition to the expression of the previously mentioned annexin genes in growth and developmental tissues, they are also expressed in response to various stress conditions. Of particular interest is that eight Hvanns are expressed in developing tillers at the six-leaf stage, with six of them being expressed in the leaf stomatal complex, which is related to stomatal development and regulation. Additionally, the other two are expressed in the leaf mesophyll, which lacks an upper and lower epidermis (Appendix A). This result may be consistent with the involvement of some of these genes in stress regulation. In the PRJNA489775, these genes were found to be expressed after NaCl treatment at different time points, indicating a close relationship between Hvanns and salt stress (Figure 3C). Meanwhile, Hvanns show expression changes under various stress conditions, suggesting a close correlation with the root stress pathway. Similar studies have been extensively conducted. HvMOREXann7 and HvMOREXann3 exhibit increased expression in stomata and after 24 h of cold stress compared to 0 h (Appendix A). HvMOREXann9 and HvMOREXann10 show increased expression in roots after 50 µM Cu stress (Appendix A). These findings collectively underscore the versatile roles of Hvanns in both the development and stress response pathways.

### 2.5. Co-Expression Network and GO Enrichment Analysis of Hvanns Involved in Tissues and under Stress Conditions

To delve deeper into the functions and molecular modules of Hvanns in different tissues and under stress conditions, we employed the Weighted Gene Co-Expression Network Analysis (WGCNA) method on RNA-seq data from various stress treatments and tissues across seven RNA-seq projects (Appendix A). This analysis unveiled 19 co-expression modules, encompassing gene counts ranging from 29 to 3459 (Appendix A). Among the 12 Hvanns investigated, 5 were identified to participate in two co-expression modules, namely, brown and yellow (Appendix A). HvMOREXann2, HvMOREXann3, HvMOREXann11, and HvMOREXann12 were grouped within the brown module, which is associated primarily with proton-transporting ATPase activity, copper ion binding, and proton transmembrane transporter activity (Figure 4B). On the other hand, HvMOREXann9 was categorized under the yellow module, characterized by blue light photoreceptor activity, protein serine/threonine kinase activity, and protein autophosphorylation (Figure 4C). Of significance, HvMOREXann3 and HvMOREXann2 displayed co-expression with eight genes (Figure 4A). Notably, among these genes, CESA6, involved in cellulose synthesis, has been reported to play a pivotal role in salt stress tolerance in Arabidopsis [29]. Additionally, RabA2b overexpression has been shown to modify the plasma membrane proteome and enhance drought tolerance in Arabidopsis [30]. Furthermore, HORVU.MOREX.r3.1HG0043600, identified as a UDP-xylose transporter, regulates Na+ ion toxicity tolerance under salt stress by interacting with OsCATs in rice [31]. In the co-expression network of HvMOREXann9, KAT2 was identified to modulate ABA (abscisic acid) responses during germination and early seedling development in Arabidopsis [32]. In essence, Hvanns exhibit substantial roles in ion transport and possess critical functions during stress responses.

### 2.6. Selective Sweep Analysis and Haplotype Analysis of Hvanns

To perform a selective sweep analysis of Hvanns during the domestication process, resequencing data from a population containing 88 wild barley samples and 194 landraces were utilized (Appendix A). The single-nucleotide polymorphism (SNP) calling pipeline generated approximately 17 million high-quality SNPs, enabling the identification of Hvanns under domestication selection. During domestication selection, based on the Fst values, the range for the top 5% was 0.43–0.81, while the range for the top 10% began from 0.38. Genes showing high selection signals within this range included HvMOREXann7, HvMOREXann8, and HvMOREXann10, with selection signals of 0.40, 0.40, and 0.48, respectively (Figure 5A). Additionally, in the analysis of π_wild_/π_doms_ values, the thresholds for the top 5% and top 10% were 4.33 and 3, respectively. Genes identified with selection signals included HvMOREXann2, HvMOREXann3, and HvMOREXann10, with ratios of 4.44, 4.44, and 3.26, respectively (Figure 5B). Notably, HvMOREXann10 was implicated in both types of domestication selection. Furthermore, through a haplotype analysis of SNP genetic variation information in the selected gene, significant differences in the haplotype frequencies of five genes were observed between wild barley and cultivated barley (Figure 5C). This finding also suggests the potential presence of a genetic bottleneck effect in the Hvanns family during the domestication transition of wild barley into landraces.

### 2.7. Validation of the Expression of Hvanns by qRT-PCR Assays

Plant annexin proteins possess diverse activities, including the regulation of Ca^2+^ channels, peroxidase functions, and ATPase/GTPase activities. These functionalities enable them to effectively respond to environmental drought stress. Transcriptome data revealed a notable upregulation of HvMOREXann7 under various stress conditions, such as salt, drought, cold, and heavy metal stresses. In this study, we aimed to investigate the potential involvement of the Hvanns gene in drought stress by evaluating the expression of HvMOREXann7 in roots under PEG-6000 treatment using qRT-PCR experimental techniques with 16 selected samples. Our experimental results indicated a significant increase in the expression of HvMOREXann7 after 6 h of PEG-6000 treatment in the wild barley samples W2, W4, W5, and W7, while no substantial change was observed in cultivated barley samples (Figure 6A,B). Subsequent analysis revealed a strong correlation (r = 0.45) between the relative expression levels of HvMOREXann7 under PEG-6000 treatment and the drought tolerance coefficient, suggesting a close association between HvMOREXann7 and drought tolerance (Figure 6D). Furthermore, significant differences in the relative expression levels of the roots were observed between the wild and cultivated populations under PEG-6000 treatment. This observation was consistent with the drought tolerance coefficient obtained from root surface area measurements (Figure 6C), providing validation of the expression levels in the selection analysis and indicating that HvMOREXann7 underwent domestication selection, playing a crucial role in drought tolerance. In conclusion, these results emphasize the importance of HvMOREXann7 in enhancing drought tolerance and highlight its potential as a key factor in the domestication process. Finally, these findings underscore the critical role of HvMOREXann7 in improving drought tolerance in barley varieties.

## 3. Discussion

Climate change, as a major contributor to both biotic and abiotic stressors, establishes a strong intrinsic link with agriculture, leading to adverse impacts on agricultural production within a given region. Climate change affects agriculture in various ways, such as changes in rainfall, fluctuations in soil temperature, pests, atmospheric ozone, and CO_2_ levels, and the melting of glaciers [33]. These changes have a negative impact on global crop production, posing a serious threat to global food security. Barley is the world’s fourth most important cereal crop, after wheat, rice, and corn. It is renowned for its beneficial effects against degenerative diseases such as diabetes, obesity, hypertension, and colon inflammation, which are often linked to dietary habits and improper lifestyles [34,35]. Additionally, barley has a wide distribution, abundant wild resources, and can adapt to various ecological environments [36]. With its rich biodiversity, exploring excellent stress-resistant genes in barley will provide beneficial genetic resources for the genetic improvement and breeding of stress resistance in staple crops such as rice and wheat, contributing to the maintenance of food security.

Plant annexins, comprising a multigene family of calcium-dependent phospholipid-binding proteins, play a vital role in responding to environmental stresses and signaling pathways during the growth and development of plants. The role of annexin genes in plant abiotic stress has been extensively studied in various crops, such as Arabidopsis [37], durum wheat [38], pepper [39], cotton [40], maize [41], and rice [42]. Annexins have been shown to regulate the flow of Ca^2+^ in the root epidermis, enhancing tolerance to salt, alkali, and osmotic stress. Additionally, they function through processes like chlorophyll degradation, the accumulation of reactive oxygen species (ROS), and a reduction in antioxidant defense capacity. Furthermore, studies have indicated their regulatory roles under high-temperature, heavy metal, and low-temperature stress conditions.

The results from the expression profiles and cis-element analysis of Hvanns indicate a close association of the annexin gene with various types of stress. In this study, we selected the gene HvMOREXann7, which has undergone domestication selection, for drought stress experiments. The results showed that the relative abundance of this gene increased in individuals with high drought resistance, while in individuals with low drought resistance, the relative abundance of this gene did not show a significant increase. Furthermore, the study indicated that, to a certain extent, wild barley exhibited a higher drought tolerance compared to cultivated varieties, which is consistent with our general understanding that wild materials possess a stronger environmental adaptability.

In this experiment, out of the 12 Hvanns obtained, 5 genes were found to exhibit signals of domestication selection. The genes HvMOREXann2, HvMOREXann3, HvMOREXann10, HvMOREXann7, and HvMOREXann8 can be identified in domestication selection signals, and whether they are involved in the domestication process can also be identified by the differentiation of the main haplotypes in wild and landrace populations. As shown in Figure 5C, the Hap4 (yellow) haplotype in HvMOREXann2, HvMOREXann3, and HvMOREXann8 was mostly found in wild populations and was almost absent in landrace populations, while the Hap3 (gray) haplotype was predominantly present in landrace populations, suggesting that Hap4 was lost during the domestication process, while Hap3 appeared to be newly generated and widely retained during domestication. Hap1, the most abundant haplotype in each gene, was present in both wild and cultivated populations, but showed significant differentiation with large differences in abundance. From the perspective of haplotypes, this also reflects that annexin genes play a certain role in the domestication process. Additionally, descriptions of their gene haplotypes revealed differences in the major haplotypes between wild and domesticated populations, demonstrating the impact of human domestication selection on barley Hvanns. Influenced by genetic bottlenecks, haplotype differentiation occurred. The lack of specific individual phenotypic traits makes it difficult to trace the distribution patterns of their haplotypes.

Annexins have been continuously investigated in both biotic and abiotic stress scenarios. For instance, in Arabidopsis, AtANN8 plays a crucial role in various stress signaling pathways [43], negatively regulating rpw8.1-mediated powdery mildew resistance and cell death, thereby linking the function of annexins to plant immunity. The knockout of the rice gene OsAnn5 renders seedlings more sensitive to cold treatments [44], while the wheat gene AtANN1 acts downstream of OST1 in the cold stress response, triggering a cascade of Ca^2+^ signals that enhances freezing tolerance [19]. Furthermore, the durum wheat annexin TdAnn6, when overexpressed in Arabidopsis, exhibits heightened promoter activity when seedlings are exposed to NaCl, mannitol, ABA, GA, and cold conditions [38]. This tissue-specific expression and cross-talk between exogenous stress stimuli suggest an additional regulatory layer for crop salt and osmotic stress responses. Overall, the involvement of annexins in various stress responses underscores their multifaceted roles in plant adaptation and defense mechanisms.

## 4. Materials and Methods

### 4.1. Identification of Annexin Genes in Barley

We selected a total of 12 barley genomes for the identification of annexin genes. This set included four wild types, four landraces, and four cultivars (Table 1). Out of these, eight barley genome sequences were obtained from the published barley pan-genome [21]. The Morex genome sequences were sourced from the Ensembl plant database website (http://plants.ensembl.org/Hordeum_vulgare/Info/Index, accessed on 12 March 2021). Three wild barley genome sequences were obtained from published articles [45,46,47]. The protein sequences were utilized for a BLASTP search (with an E-value of 1 × 10^−5^). Then, the primary annexin domain (PF00191) was obtained from the PFAM database website, which was then searched against the protein sequences of these 12 barley genomes using HMMER v3.3.2 [48], with default parameters and a filtering threshold of 0.01. After removing redundancy from the obtained protein sequences above, the integrity of the annexin conserved domain was checked using the NCBI-CDD website. The final protein’s key molecular characteristics were studied, such as the molecular weight (MW), theoretical isoelectric point (pI), and grand average of hydropathicity (GRAVY), as well as subcellular localization using the online tools ExPASy (http://web.expasy.org/protparam/, accessed on 1 March 2022) and TargetP (https://services.healthtech.dtu.dk/services/TargetP-2.0/, accessed on 5 September 2019).

### 4.2. Analysis of Evolutionary Relationships and Cis-Acting Elements

Multiple sequence alignments were performed using the ClustalW tool to analyze the evolutionary relationships of the barley annexin genes, as well as those of rice and wheat. Subsequently, Using the Maximum Likelihood (ML) model prediction tool “Find Best DNA/Protein Models (ML)” in MEGA-X (Version 11) software, the Akaike Information Criterion, AICc, and Bayesian Information Criterion, BIC, were calculated for various models and parameters to construct an ML tree. The model and parameters with the lowest AICc/BIC values were selected to build the tree. Finally, the final model JTT+G was used to construct a Maximum Likelihood tree (Appendix A). The physical chromosome locations of the barley annexin genes were displayed using the online tool MapGene2Chromosome v2.0. The identification of conserved motifs was conducted using MEME (Version 5.4.1) [49], gene structure information was obtained from the genome annotation files, and finally, visualization was performed using TBtools (Version 2.030) [50]. Gene regions and 2 kbp upstream sequences of the genes were extracted, and the PlantCARE (http://bioinformatics.psb.ugent.be/webtools/plantcare/html/, accessed on 12 March 2022) database online tool was utilized for prediction.

### 4.3. Expression Profile and Co-Expression Network Analysis of Hvanns Based on RNA-seq Data 

The Barley Expression Database (http://barleyexp.com/common.html, accessed on 31 March 2023) [51] provides RNA-seq data for nearly all barley tissues and the majority of stress experiments (Appendix A). The expression profile analysis included projects such as PRJEB14349, PRJNA704034, PRJNA489775, PRJNA767196, PRJNA382490, PRJNA752285, and PRJEB21740. FPKM values were visualized using R packages pheatmap (with the parameter scale = “row”). Additionally, a co-expression network was constructed using a collection of 7 RNA-seq projects for the expression profile analysis, covering PRJEB14349, PRJNA704034, PRJNA752285, PRJEB21740, PRJNA489775, PRJNA767196, and PRJNA382490. The Weighted Gene Co-expression Network Analysis (WGCNA) was performed using the TBtools (Version 2.030) [50] software. After obtaining the main modules of Hvanns, the corresponding modules were subjected to a Gene Ontology (GO) enrichment analysis. The GO dataset was obtained using KOBAS (http://bioinfo.org/kobas, accessed on 2 July 2021), and R packages were utilized for visualization.

### 4.4. qRT-PCR Validation

Sixteen different materials, including wild types and cultivated varieties, were used for qRT-PCR analysis (Appendix A). The seeds were disinfected with a 5% sodium hypochlorite solution for 10 min, then placed in petri dishes containing moist filter paper, and incubated in a growth chamber at a temperature of 25 °C. When the seedlings grew to a height of 2–3 cm, they were transplanted into containers with Hogland nutrient solution and maintained for 10 days, with the medium being changed every 48 h. Subsequently, the plants were cultured in a 20% polyethylene glycol (PEG6000) solution for 6 h before collecting root samples. Seedlings under normal conditions were used as controls. The roots were collected with three biological replicates for each sample. The total RNA from these samples was extracted using a plant RNA extraction kit (Shanghai Promefa, Shanghai, China) following the manufacturer’s instructions. cDNA was generated using the Reverse Transcription Kit from Accurate Biotechnology for the qPCR experiments conducted with the qPCR Kit (Accurate Biotechnology, Changsha, China). The primers used are listed in Appendix A. The qPCR program was run on a QuantStudioTM 7 Flex System (Applied Biosystems, Foster City, CA, USA), with the amplification reaction initiating at 95 °C for an initial denaturation of 30 s, followed by 40 cycles of denaturation at 95 °C for 10 s and annealing/extension at 60 °C for 30 s. Three technical replicates were included in this experiment, and the expression levels were calculated using the 2^−ΔΔCT^ method.

### 4.5. The Drought-Tolerant Coefficient for Barley Drought Tolerance

The materials mentioned above were used to evaluate drought resistance. The hydroponic cultivation method followed the provided guidelines. Starting at the three-leaf stage, a 20% polyethylene glycol (PEG6000) solution was applied. Both the treatment and control groups were subjected to the same conditions, with the stress treatment lasting 20 days. The barley roots were scanned using aflatbed scanner (Epson Perfection V700 Photo, Epson China, Shanghai, China), and the WinRHIZO (Regent Instruments, Quebec, QC, Canada) root analysis system was used for an image analysis to collect data on the root surface area. The drought-tolerant coefficient was evaluated using the ratio of the root surface area after PEG-6000 treatment to the control [52].

### 4.6. Selective Sweep Analysis and Haplotype Analysis of Hvanns

Utilizing low-depth whole-genome shotgun sequencing from over 282 diverse and widely distributed population materials (the information for all variant sites was derived from low-coverage resequencing data, which included 88 wild and 194 landrace accessions), a selective sweep analysis was conducted. The data were sourced from the published barley pan-genome research, specifically, the whole-genome shotgun sequencing project with the identifier PRJEB36577, with sample information listed in Appendix A. The clean reads were aligned to the barley reference genome Morex V3 using BWA-MEM 0.7.17-r1188. The HaplotypeCaller tool in GATK (version 4.2.0.0) was used to generate and call single-nucleotide polymorphisms (SNPs), and then the SelectVariants option was utilized to filter and separate the identified SNPs. After generating a Variant Call Format (VCF) file using the Visual Component Framework, PLINK (version 1.90b6.21) was used to filter the SNPs with a minor allele frequency (maf) of 0.05 and a genotyping rate (geno) of 0.1. Then, the fixation index (Fst) and nucleotide diversity (π) were analyzed using vcftools (version0.1.16) with a window size of 25 k. The results were then visualized using the R package CMplot (version 3.6.0). Performing a haplotype analysis on the selected gene region and the upstream 2 k region using DnaSP (version 6) [53] will provide valuable insights into the genetic variations within these regions and the distribution of haplotypes.

## Figures and Tables

**Figure 1 ijms-25-03883-f001:**
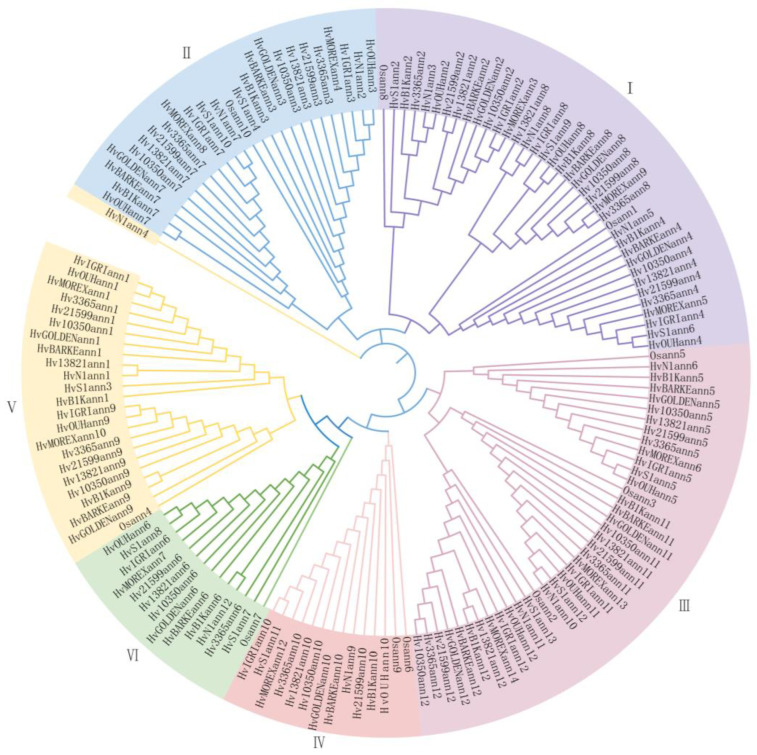
A Maximum Likelihood (ML) tree for annexin genes in *Hordeum vulgare* (Hv), Triticum aestivum (Ta), and Oryza satival (Os). The background colors represent Hvanns subgroups, which are labelled in the out layer.

**Figure 2 ijms-25-03883-f002:**
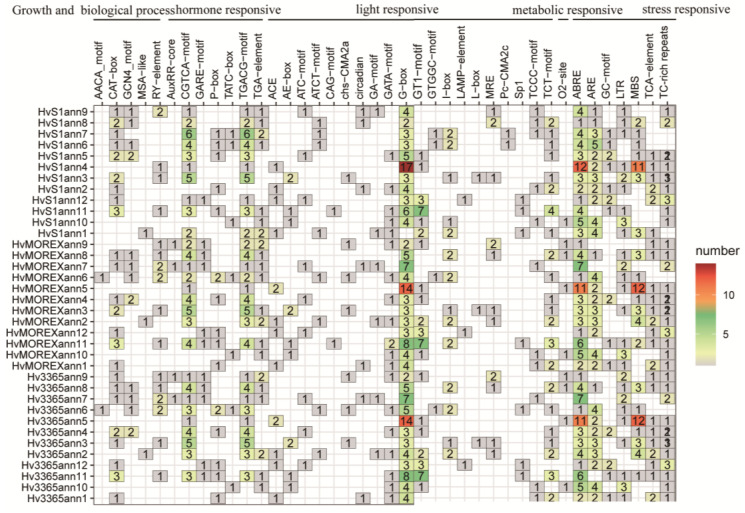
The number and function classification of cis-acting element for the genome EC-S1, Morex, and HOR3365.

**Figure 3 ijms-25-03883-f003:**
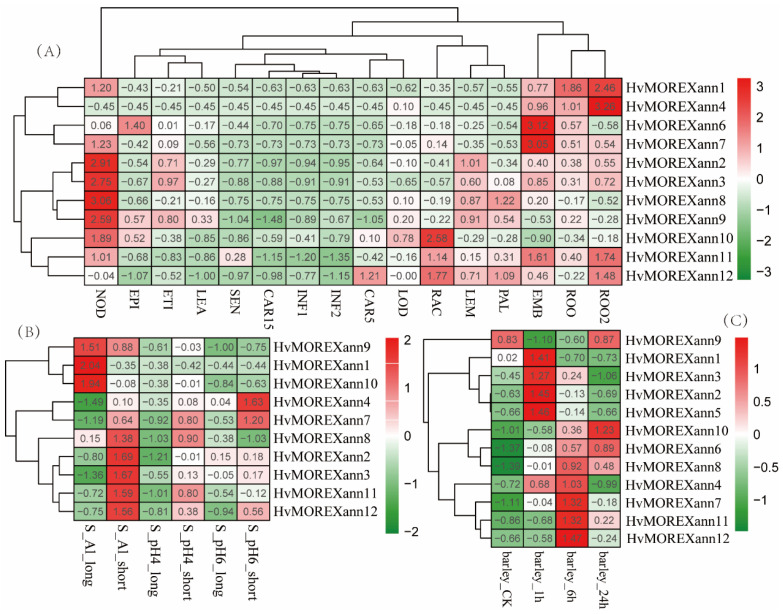
Expression profiles of Hvanns in different tissues and under diverse stresses. (**A**): Expression profiles of 16 developmental stages from PRJEB14349. Bracts removed grains at 15DAP(CAR15); bracts removed grains at 5DAP (CAR5); embryos dissected from 4-day-old germinating grains (EMB); epidermis at 4 weeks old (EPI); etiolated from 10-day-old seedling (ETI); young inflorescences with 5 mm (INF1); young inflorescences with 1–1.5 cm (INF2); shoot with the size of 10 cm from the seedlings (LEA); lemma at 6 weeks after anthesis (LEM); lodicule at 6 weeks after anthesis (LOD); developing tillers at the six-leaf stage (PAL); 6-week-old palea (NOD); rachis at 5 weeks after anthesis (RAC); root from 4-week-old seedlings (ROO2); and roots from the seedlings at the 10 cm shoot stage (ROO). (**B**): Transcriptome analysis of barley root meristem under Al and low pH stress, from PRJNA704034. Aluminum treatment at low pH (4.0) (S_Al_longzZ); aluminum treatment at low pH (4.0) (S_Al_short); low pH (4.0) (S_pH4_long); low pH (4.0) (S_pH4_short); optimal pH (6.0) (S_pH6_long); and optimal pH (6.0) (S_pH6_short). (**C**): Transcriptome analysis of salt-treated roots from PRJNA489775. Normal condition (barley_CK); salt treatment for 1 h (barley_1 h); salt treatment for 6 h (barley_6 h); and salt treatment for 24 h (barley_24 h). FPKM values are visualized using R packages pheatmap (with the parameter scale = “row”). The detailed content can be found in Appendix A.

**Figure 4 ijms-25-03883-f004:**
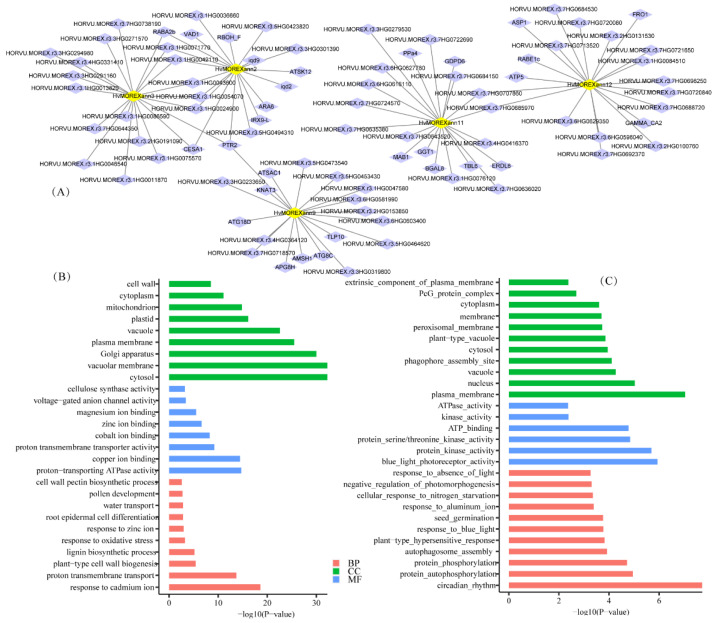
Co-expression network based on transcriptome data under normal growth conditions and stress conditions by the WGCNA method (**A**). GO enrichment analysis of co-expression modules brown (**B**) and yellow (**C**).

**Figure 5 ijms-25-03883-f005:**
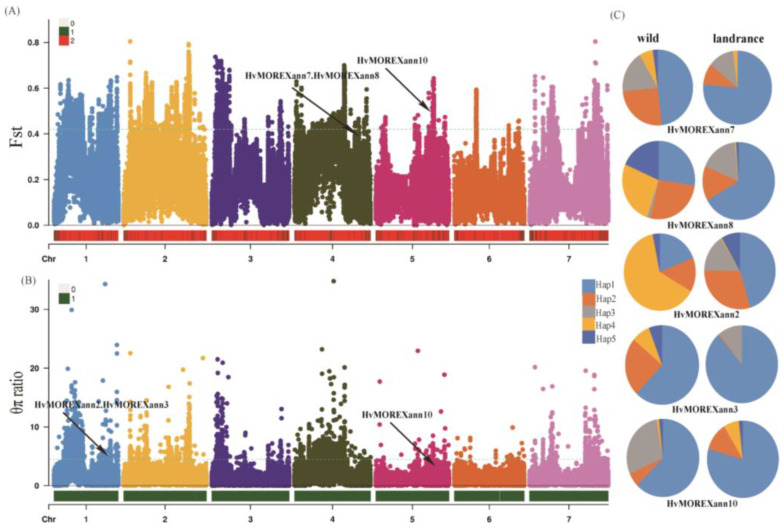
Selective sweep analysis and haplotype analysis based on resequencing data. (**A**) The fixation index (Fst) and πwild/πdoms values (**B**) between landrace barley and wild barley populations; Green dashed lines represent the top 5% threshold. (**C**) Haplotype frequency analysis of Hvanns in landrace barley and wild barley populations.

**Figure 6 ijms-25-03883-f006:**
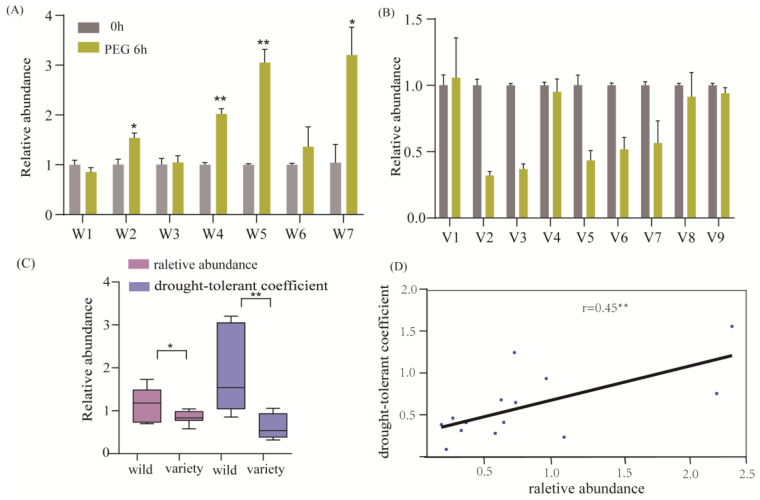
Relative expression levels of HvMOREXann7 under drought stress conditions using the qRT-PCR method. (**A**,**B**): Relative expression levels in roots of seven wild barley and nine barley varieties under PEG-6000 treatment. Data represent the average of three technical replications and A bar chart with the mean value including the standard deviation. (**C**): Differences in the relative expression levels and drought tolerance coefficient between wild barley and barley varieties. The statistical method used is a two-tailed *t*-test. (* *p* < 0.05, ** *p* < 0.01). (**D**): Correlation analysis of the drought tolerance coefficient and relative expression levels (average of three technical replications) in roots.

**Table 1 ijms-25-03883-t001:** The count of Hvanns and their characteristics and biochemical properties.

Type	Accessions	Amino Acids	pI	GRAVY	Cytoplasm	Mitochondria	Nucleus Cytoplasm	Nucleus	Chloroplast
Wild	B1K-04-12	334.42	7.72	−0.35	7	1	1	1	2
	EC-N1	330.33	8.01	−0.42	7	0	0	2	3
	EC-S1	304.33	7.7	−0.31	6	1	0	1	4
	OUH602	334.42	7.77	−0.35	6	1	1	1	3
Landrace	HOR10350	334.42	7.71	−0.35	6	1	1	1	3
	HOR13821	334.42	7.72	−0.36	6	1	1	1	3
	HOR21599	334.42	7.73	−0.36	6	1	1	1	3
	HOR3365	334.25	7.72	−0.35	6	1	1	1	3
Variety	IGRI	334.58	7.72	−0.35	6	1	1	1	3
	Morex V3	329.36	7.73	−0.36	6	1	1	1	3
	Barke	329.67	7.68	−0.35	6	1	1	1	3
	Golden	329.67	7.68	−0.35	6	1	1	1	3

## Data Availability

All of the datasets supporting the results of this article are included within the article and its additional files.

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
