# Peer review of "The Evolution, Expression Patterns, and Domestication Selection Analysis of the Annexin Gene Family in the Barley Pan-Genome"

_ijms, 2024, doi:10.3390/ijms25073883_

Round 1

Reviewer 1 Report

Comments and Suggestions for Authors

In this manuscript, Chen et al. comprehensively resolved the annexin gene family in 12 barley (Hordeum vulgare L.) reference genomes, including cultivars, landraces, and wild barley. To further investigate the evolution and expression patterns of Hvanns family, Chen et al. employed phylogenetic analysis, cis-acting element analysis, and transcriptome analysis on Hvanns. The manuscript presents results indicating that Hvanns are involved in growth, development and response to various stresses. Additionally, through selective sweep analysis, haplotype analysis and validation of the expression of Hvanns, HvMOREXann7 and HvMOREXann10 were identified as domestication-selected genes that play crucial roles in salt and drought tolerance. The manuscript offers a detailed examination of annexin gene family in barley, providing valuable insights for barley breeding in extreme climates.However, there are several points that were unclear to me and require further clarification.

Major comments

1.      Cis-Element analysis (Lines 170-195): it is unclear what is meant by 'group 5 Hvanns' in fig2. The previous group is divided according to the phylogenetic tree of protein sequences, and its result does not prove that Annexin genes with the same ID number in different assemblies are in the same group. For example, HvMOREXann5 is in group III while HvN1ann5 belongs to group I. Therefore, it is not appropriate to use group 5 directly in this context. What is the difference between the 'G-box' and 'G-Box' in Figure 2? Additionally, the last column in Figure 2 labelled 'NA' is not explained in the legend or supplementary tables, and some columns contain overlapping numbers. The authors should clarify these points. The available evidence does not support the conclusion that annexin and seed development are related. The presence of fewer O2-sites in wild varieties compared to landraces and varieties is not a direct indication of this relationship. Additionally, the relationship between AACA_motif and seed development is not well described.

2.      Lack of necessary statistical analyses and figure notes: The conclusions drawn in the text are not supported by the necessary statistics. For instance, in Line 154, it is stated that there is a strong correlation between gene structures, conserved motifs, phylogeny, and subcellular localization. However, the correlation between subcellular localization and the other three is not reflected in Figure S2, and the data given in Table S2 are not intuitive enough. S2, and the data provided in Table S2 is not intuitive enough. Line 184, Figure 2 also does not visually demonstrate that the amount of O2 site element was lower in wild populations compared to landraces and varieties. The legend in Figure 3A should provide specific information about the different treatments. The specific treatments corresponding to Figure 3B and Figure 3C should be labelled in the figure.

3.      Data used in this study: To ensure the study's conclusions are more robust, it is recommended to include all published barley genomes, which currently number more than 12. The source of the barley population resequencing data used in the analysis in Figure 5, whether newly generated for this study or publicly available, should be stated. One aspect of qRT-PCR validation that is puzzling is the difference in relative expression levels between the samples used in Figure 6A and Figure 6B, which should be the same samples, but under the same treatment with PEG 6h, the trend of change in relative expression levels changes, i.e., the relative expression levels of "W5" decreased in Figure 6A, whereas they increased significantly in Figure 6B. For instance, the relative expression levels of 'W5' in Figure 6A have increased significantly, while in Figure 6B they have decreased. Additionally, the sample names in Figure 6B and Figure 6A are inconsistent, as are those in Table S10.

Other minor comments.

Line 36: A full stop should be added between “process” and “Finally”. Therefore, it should look like this: “...domestication process. Finally, qRT-PCR analysis confirmed...”

Lines 79-80: Capitalization of a sentence's initial letter is required.

Line 158: There is no evidence to suggest that the insertion of non-LTR retrotransposons is the cause.

Line 186: The conclusion that the AACA_motif element appears in four Hvanns does not correspond to Figure 2 or the results of schedule Table S3, which indicate three.

Line 275: Change "0.383" to "0.38".

Line 288: According to the information presented in the paper, it is not appropriate to use Figure 6D as a line graph. Additionally, it appears to be partially redundant with Figure 6C.

The quality of the Figure S1 is hard for reading the information. It is impossible to distinguish between assemblies and chromosomes.

Figre S2 lacks a legend for the Figure S2D.

“Foma_spikes” and “Deficiens2_spikes” in Figure S3 have no counterpart in the Table S7.

Author Response

point-by-point response

Reviewer 2 Report

Comments and Suggestions for Authors

I have had the opportunity of reviewing the manuscript titled: “Evolution, Expression Patterns, and Domestication Selection Analysis of the Annexin Gene Family in the Barley Pan-Genome“, and I am able to provide my assessment. The study's focus on the impact of climate change on barley and the potential implications for global food security.

Overall, the study is well executed, contributing valuable insights into the role of annexin genes in barley under stress conditions. The methods employed, includes the identification of annexin genes, evolutionary relationship analysis, and expression profile examination. The manuscript is well-organized, with a logical flow from introduction to conclusion.

Nevertheless, I believe some improvements are necessary, for the manuscript to be suitable for publication:

The methods would benefit from a more detailed description in some parts. Include more explicit details about the statistical analyses performed. Mention the specific tests used and the criteria for significance. What was the model used for the construction of the neighbor-joining tree (line 379), and how was this model chosen?

To help readers understand the basis for the work conclusions, it would be helpful to provide additional details on the criteria  used for evaluating drought resistance in the materials evaluated in the work.

In the Results section, it would be helpful to provide brief reminders or references to the methodologies described in the Methods at the beggining of each results section.

Some sections of the results could benefit from additional clarification. For instance, in the discussion of the selective sweep analysis, a more detailed explanation of the observed patterns and their implications would enhance the reader's understanding.

Ensure consistent use of terminology throughout the manuscript. For instance, if specific terms or abbreviations are introduced in the Methods section, make sure they are consistently used in the Results and Discussion sections. This also applies to species names. For example, in lines 334-336, authors use scientific names and common names without any criteria whatsoever: “Arabidopsis thaliana[37], durum wheat[38], Capsicum cannuum[39], cotton[40], Zea mays[41], and rice[42]”

The authors may consider placing their findings in a broader context. How do their results align with or contribute to existing literature on annexin genes in other crops under stress conditions? Also, including a brief discussion on potential avenues for future research based on the current findings would add depth to the conclusion.

In conclusion, in my opinion, the manuscript is suitable for publication, pending the incorporation of the suggested revisions.

Round 2

Reviewer 2 Report

Comments and Suggestions for Authors

After the revision provided by the authors, with considerable improvements, I believe the manuscript is suitable for publication